

# $\mathbb{Z}_4$ transitions in quantum loop models on a zig-zag ladder

**Bowy M. La Rivière⋆ and Natalia Chepiga**

Kavli Institute of Nanoscience, Delft University of Technology,
Lorentzweg 1, 2628 CJ Delft, the Netherlands

⋆ b.m.lariviere@tudelft.nl

## Abstract

We study the nature of quantum phase transitions out of $\mathbb{Z}_4$ ordered phases in quantum loop models on a zig-zag ladder. We report very rich critical behavior that includes a pair of Ising transitions, a multi-critical Ashkin-Teller point and a remarkably extended interval of a chiral transition. Although plaquette states turn out to be essential to realize chiral transitions, we demonstrate that critical regimes can be manipulated by deforming the model as to increase the presence of leg-dimerized states. This can be done to the point where the chiral transition turns into first order, we argue that this is associated with the emergence of a critical end point.

| | |
|---|---|
| Received | 04-07-2024 |
| Accepted | 21-10-2024 |
| Published | 26-11-2024 |

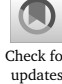
# 1  Introduction

Identifying different universality classes of quantum phase transitions appearing in low dimensional many-body systems is one of the central topics in condensed matter physics [1]. Often, one can guess the underlying critical theory by analyzing the symmetry that is spontaneously broken at the transition. Let us consider, for example, the transition between an ordered phase with a $\mathbb{Z}_p$ symmetry and a disordered phase. Naïvely, the corresponding transition is expected to belong to the universality class of conformal minimal model with the matching value of $p$, i.e. Ising for $p = 2$, three-state Potts for $p = 3$, Ashkin-Teller for $p = 4$. However, if short-range correlations in the disordered phase are incommensurate (IC), with the dominant wave-vector $q$ different from its commensurate value $2\pi/p$, the transition is classified as a commensurate-incommensurate (C-IC) one. Understanding the nature of these C-IC transitions is one of the biggest challenges of modern quantum physics that roots back to the classical study of absorbed monolayers [2–5]. In this case, different sequences of ground-state domains, such as *ABC* and *ACB*, have different sets of domain walls, and if free energy contributions of various domain walls are not identical, such as $AB \neq AC$, one type of sequence becomes energetically favored over the other and induces chiral perturbations [6]. For $p = 2$, if the transition is continuous, one can always expect the transition to be of the Ising type, irregardless of chiral perturbations. But for $p > 2$ chiral perturbations are (almost always) relevant and drastically change the nature of the transition [3, 6–8]. For instance, for ordered phases with periodicities $p \geq 5$ the C-IC transition cannot be direct but is a two-step transition with an intermediate floating phase [3, 9], i.e. an incommensurate Luttinger liquid [10]. This floating phase is separated from the C ordered phase by a Pokrovsky-Talapov [2, 11] transition and from the IC disordered phase by the Kosterlitz-Thouless transition [12], with exponentially decaying correlation lengths for the latter.

The transitions in the cases of $p = 3$ and $p = 4$ are much more exotic in nature. When the disordered phase is commensurate, the transition is conformal and belongs to the 3-state Potts (for $p = 3$) or Ashkin-Teller (for $p = 4$) universality class. In the presence of strong chiral perturbations the transition to the $p = 3$ and $p = 4$ phases is through the floating phase, similar to the $p \geq 5$ cases. In the presence of weak chiral perturbations the C-IC transition to the $p = 3$ phase remains direct, though it is no longer conformal but belongs to the chiral Huse-Fisher universality class [3, 6, 13–17]. Weak chiral perturbations might also lead to a direct chiral transition at the boundary of the $p = 4$ phase. In this case, however, the appearance of the chiral transition is a subtle issue [3] due to the fact that the Ashkin-Teller family of conformal critical theories forms a so-called *weak* universality class [18, 19]. This means that some critical exponents, such as $\nu$ and $\beta$ describing the divergence of the correlation length and the scaling of the order parameter in the vicinity of the critical point respectively, are not fixed but vary as a function of an external parameter (usually called $\lambda$ [18, 20–22]), while others, including the central charge and the scaling dimension $d = \beta/\nu$, are universal within the family. The Ashkin-Teller family of transitions ranges from a pair of decoupled Ising chains with $\nu = 1$ to the symmetric 4-state Potts point with $\nu = 2/3$ [18]. It turns out that the way chiral perturbations affect the nature of the transition drastically depend on the properties of the conformal point. When $\nu \gtrsim 0.8$ even weak chiral perturbations always open up a

floating phase [3, 20, 22]. On the other hand, when the Ashkin-Teller point is characterized by $(1 + \sqrt{3})/4 < \nu \lesssim 0.8$ a direct chiral transition emerges under weak chiral perturbations and is followed by an opening of the floating phase when chiral perturbations become strong. The two regimes are separated by the Lifshitz point that is characterized by a dynamical critical exponent $z = 3$ [3, 16, 20]. Finally, when the Ashkin-Teller point is close to the 4-state Potts point with $2/3 \leq \nu \leq (1 + \sqrt{3})/4 \approx 0.683$, chiral perturbations are irrelevant [2], allowing for an interval of a conformal Ashkin-Teller transition, followed by a chiral transition and the floating phase when chiral perturbations become stronger [22].

In addition to the chiral Ashkin-Teller model [20, 22], the $\mathbb{Z}_4$ chiral transition has been recently reported in the context of Rydberg atoms [23–30]. In experiments Rydberg atoms are trapped with optical tweezers in a one-dimensional array with a well controlled inter-atomic distances. Lasers with Rabi frequency $\Omega$ bring atoms from their ground-state to excited Rydberg states. Competition between the strong van der Waals interaction of excited atoms and the laser detuning $\Delta$ leads to a rich phase diagram with the lobes of density-wave phases with different integer periodicities $p$ [9, 24]. A narrow intervals of the chiral transition has been observed at the boundary of the $p = 4$ lobe [26, 27, 29].

In this paper we explore a possibility to realize the $\mathbb{Z}_4$ chiral transition in quasi-one-dimensional quantum magnets. The transition we are looking for takes place between two gapped phases, one of which - the disorder phase - does not break translation symmetry and has no long-range order. According to the Mermin-Wagner theorem [31] this naturally exclude half-integer spin chains. Previous studies of spin-1 Heisenberg zig-zag chains with anti-ferromagnetic nearest- and next-nearest-neighbor interactions report a next-nearest-neighbor Haldane (NNN-Haldane) phase appearing when the ratio between these interactions exceeds $J_2/J_1 \gtrsim 0.75$ [32–35]. Intuitively one can understand NNN-Haldane phase by starting with two decoupled Haldane chains and tuning the antiferromagnetic nearest-neighbor coupling in such way that the spin-1/2 edge states are annihilated and a valence bond connects the two chains at each edge, forming a single spin-singlet loop spanning over the entire zig-zag ladder.[1] This phase, being topologically trivial without long-range order and with incommensurate short-range correlations, is therefore ideally suited for the role of an incommensurate disordered phase, providing an excellent starting point for our study. The $\mathbb{Z}_4$ chiral transition, just like the previously reported Ising transition to the dimerized phase [35, 36], is expected to be non-magnetic with low-lying excitations taking place entirely within the singlet sector while a singlet-triplet gap remains open.[2] This allow us to focus on $S = 1$ quantum loop models (QLMs) [37], which effectively disregard all magnetic degrees of freedom such that the Hilbert space is formed by states with all possible spin-singlet, often called dimers, coverings satisfying some quantum loop constraints. We note that $S = 1$ QLMs are rather similar to the class of quantum dimer models [38–40] that were originally proposed in the context of high temperature cuprate superconductors. These provide an effective description of valence bond crystals [41–43], think of columnar, staggered and plaquette phases on a square lattice, and resonating valence bond phases [44–47] on various lattices. The main difference between the latter and former models is that states of $S = 1$ QLMs consist of dimer coverings with two dimers per site while quantum dimer states only allow for one dimer per site. This makes it so that in QLMs, contrary to its quantum dimer model counterpart, the dimers can form loops. Often, these model do not allow trivial loops - formed only between two lattice sites. By contrast, in the present case we do allow them.

These quantum loop models provide a number of computational advantages. Firstly, QLMs are by definition constrained: at each node, originally hosting a spin-1 site, 2 and only 2

---

[1]The spin-1/2 edge states in the NNN-Haldane phase cause the transitions out of it to be magnetic in nature, making it rather different from the NNN-Haldane phase, with a topological term in the corresponding field theory.

[2]In the literature these transitions are sometimes known as valence-bond-singlet- or VBS-transitions.

quantum dimers originate. Dimer coverings that violate this constraint are excluded from the Hilbert space, significantly reducing its size. Secondly, QLMs formulated directly in terms of dimers simplifies the construction of the parent Hamiltonian that would lead to a desired long-range order with spontaneously broken $\mathbb{Z}_4$ symmetry. Furthermore, there are several realization of the $\mathbb{Z}_4$ phase in spin-1 chains and, as we will show later, quantum loop models provide a natural way to interpolate and distinguish between them. Thirdly, in special cases QLMs can be mapped to the blockade models of Rydberg atoms [48]. This provides an excellent starting point for our study and opens a by-path for experimental validation of our results. Fourthly, the non-magnetic Ising transition into the dimerized phase in a frustrated Haldane chain mentioned previously has been successfully reproduced with the related quantum loop model already [48]. In other words, if a chiral transition exists in quantum spin-1 chains, it will likely also appear in quantum loop models. In turn, realization of chiral transitions in QLMs will help to narrow down the conditions for its appearance in more realistic models of quantum magnetism. We address the problem numerically with the state-of-the-art density matrix renormalization group (DMRG) algorithm [49,50] with explicitly implemented quantum loop constraints [48] such that the algorithm fully profits from a restricted Hilbert space, allowing us to reach convergence for critical chains with up to $N = 3000$ sites.

In this paper we report an extremely extended chiral transition that separates the $\mathbb{Z}_4$ leg-dimerized and the disordered NNN-Haldane phase. For a certain parameter range we see that the disordered and $\mathbb{Z}_4$ phase are connected via a pair of Ising transitions with a $\mathbb{Z}_2$ rung-dimerized phase between the two. Eventually the rung-dimerized phase disappears and two Ising transitions merge into a multi-critical Ashkin-Teller point, beyond which we resolve the $\mathbb{Z}_4$ chiral transition. We show that the nature of the Ashkin-Teller point and the extent of the chiral transition depends on the relative weight of two realizations of the $\mathbb{Z}_4$ phase - trivial double-dimers on legs and four-site plaquette loops. Furthermore, and quite surprisingly, when the weight of double leg dimers is too high the chiral transition turns into a first order transition. We argue that this might be due to the presence of an additional relevant operator in the underlying critical theory.

The rest of the paper is organized as follows. In Section 2 we define the model, provide details on our DMRG algorithm and explain how we extract main observables. In Section 3 we discuss the basic phase diagram of the quantum loop ladder with three main phases - $\mathbb{Z}_2$, $\mathbb{Z}_4$ and disordered. We also show that for an extremely extended parameter range the transition between the latter two phases is chiral. In Section 4 we show that the nature of the multi-critical point and the extent of the chiral transition can be manipulated by the relative ratio of the two kinetic terms of the QLM. Here we also discuss how the first order transition between the $\mathbb{Z}_4$ and the disordered phase develops as a function of the kinetic term responsible for the formation of double-dimers. Finally, we summarize our results and put them in perspective in Section 5.

## 2 Model & methods

### 2.1 Model

We study $\mathbb{Z}_4$ transitions in QLMs with two dimers per node on a zig-zag ladder. This choice allows us to realize a quantum dimer analogue of the NNN-Haldane phase of the $J_1 - J_2$ Heisenberg spin-1 chain. We also include a potential term that stabilizes the $\mathbb{Z}_4$ phase. The microscopic Hamiltonian is defined as follows:

$$\mathcal{H}_{\text{QLM}} = -\sum_{\text{plaquettes}} \left( t |\diagbox\rangle\langle\diagbox| + t' |\diagbox\rangle\langle\diagbox| + \text{h.c.} \right) - \theta \sum_{\text{plaquettes}} |\diagbox\rangle\langle\diagbox| + \delta \sum_{\text{rungs}} |\diagbox\rangle\langle\diagbox|, \quad (1)$$

where the respective sums run over all plaquette and rungs with both orientations of the zig-zag chain. The first kinetic term, parametrized by $t$, acts on a plaquette that has its rungs occupied by a dimer (either single or double) and flips one pair of dimers at a time over a given plaquette. The second kinetic term controlled by $t'$ acts on a plaquette that has both its rungs and legs occupied by a single dimer and flips them to a plaquette with two dimers on both legs. Furthermore, the first potential term, controlled by the parameter $\theta$, favors 4-site quantum loops - the plaquette phase. Lastly, the parameter $\delta$ controls the second kinetic term that acts on all rungs in the ladder. This term aims to suppress the otherwise very stable rung dimer phase. Without loss of generality we set $t = 1$ and explore the model as a function of three independent parameters $t'$, $\theta$ and $\delta$.

## 2.2 DMRG

We study the model defined in Eq.(1) with the constrained three-site - optimizing three tensors per iteration - DMRG algorithm with explicitly implemented quantum loop constraints, and using a Matrix Product State (MPS) ansatz [48–52]. With all eigenstates fully consisting of local dimer configurations, the QLM degrees of freedom can be taken as the occupation of dimers on the rungs and legs instead of spins. We make use of this by considering the occupation of dimers on a single rung with the preceding leg as the local Hilbert space, with its dimension being equal to six - one state without any dimers, two with two dimers on either the rung and leg, and three with single dimer occupations of the rung and leg. For different dimer occupations, we associate quantum numbers to label their respective sectors in the Hilbert space in iterative processes of the algorithm, such as the construction of the left and right normalized blocks. These blocks are constructed iteratively and the fusion between these quantum numbers under the addition of new sites to these blocks are described by a set of fusion rules. Since there is a one-to-one correspondence between quantum labels, the tensor can be projected into a block-diagonal form, significantly reducing computational costs. In addition, the dimension of the Hilbert space scales $\approx 1.68^N$ instead of $3^N$ for spin-1 chains. Further details on the construction of the matrix product operator for the QLM Hamiltonian and technical aspects of the implementation of the constraints are provided in Appendix A.

We simulate systems with up to $N = 3000$ spin-1 lattice sites, which effectively translates to $N = 2999$ QLM local degrees of freedom, each half sweep we increase the bond dimension by 200, up to a maximum of 10000 and discard all singular values smaller than $10^{-8}$. We note that during simulations the maximum number of states, even for the largest system considered, was never reached. Due to fragmented Hilbert space of QLMs, tensor operations can be performed block by block and the complexity is reduced by a lot. We perform up to seven full sweeps. As a convergence criteria we require the absolute error of the energy over a sweep to be below $10^{-13}$. We start each simulation with a random guess for our initial wave-function. We always use $4k-1$, with $k \in \mathbb{N}$, QLM sites to ensure that the ordered phases, including the $\mathbb{Z}_4$ plaquette and leg-double-dimers, fully cover the ladder with symmetric boundary conditions.

## 2.3 Extraction of $\xi$ and $q$

Our numerical analysis of the nature of the transition between the $\mathbb{Z}_4$ symmetric and the NNN-Haldane phase primarily relies on the scaling of the correlation length $\xi$ and the incommensurate wave-vector $q$. We extract both of these from the correlation function $C_{i,j} \propto \langle n_i n_j \rangle - \langle n_i \rangle \langle n_j \rangle$ - unless states otherwise we use $n_i = |\nearrow\rangle\langle\nearrow| + |\nearrow\rangle\langle\nearrow|$ - by fitting it to the Ornstein-Zernicke form [53]:

$$C_{i,j}^{\text{OZ}} \propto \frac{e^{-|i-j|/\xi}}{\sqrt{|i-j|}} \cos(q|i-j|\pi + \phi_0), \tag{2}$$


where $\xi$, $q$ and the phase $\phi_0$ are considered as fitting parameters. The correlation function is always calculated on the interval $N/2 \leq i \leq N$ with $j = N/2$. We extract the correlation length and the wavevector in a two-step process. First, we plot the logarithm of the correlation function, as shown in Fig.1 (a), and fit its slope with the function $\ln C_{i,j}(x = |i-j|) \approx A - x/\xi - \ln(x)/2$ to extract $\xi$. In doing so we fit $\xi$ and the amplitude $A$ at the same time. To extract the slope we first identify local maxima of $\ln|C_{i,j}|$ by comparing each point with its 2 neighbours and then fitting these maxima.

In the second step we calculate the reduced correlation function

$$\tilde{C}_{i,j} = C_{i,j} \cdot A e^{|i-j|/\xi} \sqrt{|i-j|}, \tag{3}$$

that we fit with

$$\tilde{C}_{i,j}^{\text{OZ}} \approx a\cos(q|i-j|\pi + \phi_0). \tag{4}$$

In Fig.1 (b) we show an example of the fit where we treat $q$, $\phi$ and $a$ as fitting parameters. The agreement between our numerical data (blue circles) and the fit (orange dots) is almost perfect. We expect the wave-vector $q$ to be affected by finite-size effect due to fixed boundary conditions. We can estimate the associated error to be of the order of $2\xi/N^2$, where $2/N$ is a single step in $q$ and the extra factor $\xi/N$ results from finite size effects that could arise at the boundaries.

We repeat the procedure outlined above for multiple points in proximity to the critical point. From these data points we extract the critical exponents $\nu$ and $\bar{\beta}$ from the scaling of the correlation $\xi$ and the wave-vector $q$ respectively. This we also do in two steps. First we fit the inverse of the correlation length $1/\xi \propto |\theta - \theta_c|^\nu$ to extract $\nu$ and the critical parameter $\theta_c$, indicating the location of the transition. We do this with the function $1/\xi = A\Theta(\theta_c - \theta)(\theta_c - \theta)^\nu + B\Theta(\theta - \theta_c)(\theta - \theta_c)^\nu$, where $\Theta(x)$ is the heaviside step function and we fitted $\nu$ and $\theta_c$ but also the constants $A$ and $B$ all at the same time. After extract

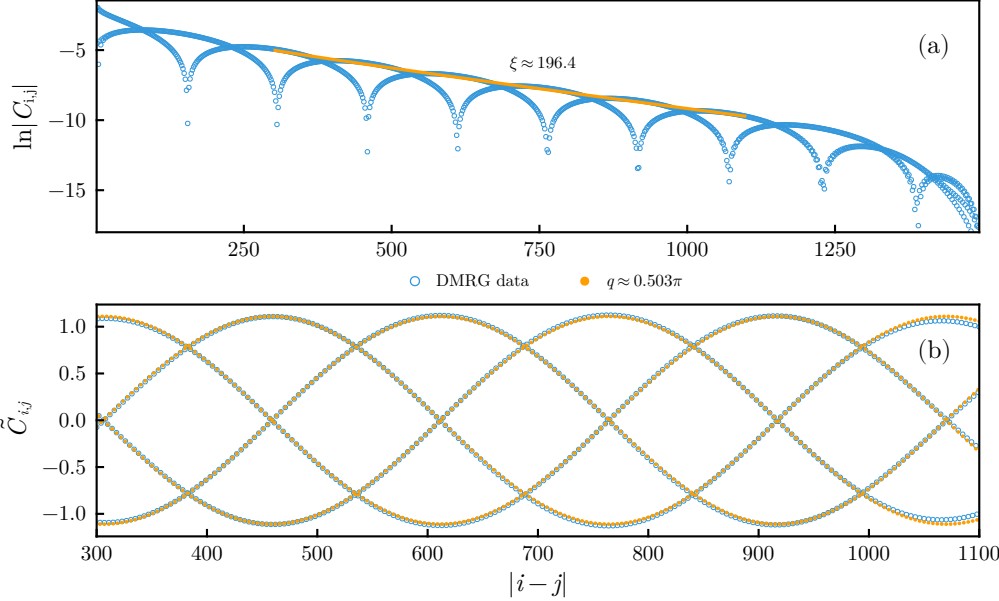

Figure 1: Example of fitting the correlation function $C_{i,j}$ to the Ornstein-Zernicke form (2) in a two step process. (a) First the correlation length $\xi$ is extracted from fitting the main slope of $\ln|C_{i,j}|$. (b) Then the reduced correlation function $\tilde{C}_{i,j}$ is calculated and fitted to extract the wavevector $q$.

$\theta_c$ we use it to fit the wavevector $q \propto |\theta - \theta_c|^{\bar{\beta}}$ with $q = C\Theta(\theta_c - \theta)(\theta_c - \theta)^{\bar{\beta}}$ to obtain $\bar{\beta}$. In our analysis we also compute the product $|q/\pi - 1/2| \times \xi$. We would like to emphasize here that we do what we outlined above point-by-point, independently for each fit for $\nu$ and $\bar{\beta}$.

# 3 $\mathbb{Z}_4$ transitions for $t' = t$

## 3.1 Overview of the phase diagram

We start our study of the quantum loop model defined in Eq. (1) by looking at a special case $t' = t = 1$. The ground-state phase diagram as a function of $\delta$ and $\theta$ is presented in Fig.2 and contains three main phases - NNN-Haldane phase, $\mathbb{Z}_2$ rung-dimerized phase and $\mathbb{Z}_4$ leg-dimerized phase. The NNN-Haldane phase is a disordered phase realized for positive $\delta$ and small $\theta$. Within the disordered phase we distinguish two regions: one with commensurate and the other with incommensurate short-range order, which are separated by the so-called disorder line. In Appendix B we explain in details the method we use to locate this line. By increasing $\theta$ while keeping $\delta$ positive, the system enters the $\mathbb{Z}_4$ leg-dimerized phase phase with spontaneously broken translation symmetry. There are two main families of states realized with a broken $\mathbb{Z}_4$ symmetry - plaquette and columnar leg states. In the former every four consecutive sites form a single dimer loop while in the latter every other leg is occupied by a double dimer - a trivial loop. Example of both states are sketched in Fig.2. The leg-dimerized phase realized with $t' = t$ is a superposition of these two families of states. The columnar states' weight is most significant near the boundary of the leg-dimerized phase, but, nonetheless, is always a fraction of that of plaquette states. Further away from the boundary

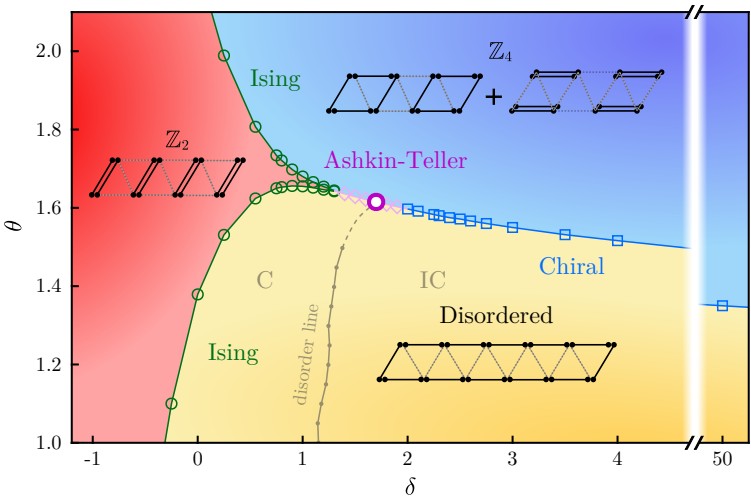

Figure 2: Phase diagram of the quantum loop model defined in Eq. (1) for $t' = t = 1$ as a function of $\delta$ and $\theta$. There are three main phases: disordered NNN-Haldane phase (yellow) with a commensurate and incommensurate part separated by the disorder line (grey line), $\mathbb{Z}_2$ rung-dimerized phase (red), and $\mathbb{Z}_4$ phase (blue) as a superposition of plaquette and columnar states. The transition between the disordered phase and the rung-dimerized phase and that between the rung-dimerized and $\mathbb{Z}_4$ leg-dimerized phase are of the Ising type (green circles). Two Ising transitions merge into the multi-critical Ashkin-Teller point (purple dot) within some uncertainty of its location indicated by the light-pink region around it. Beyond the Ashkin-Teller point the transition is chiral (blue squares) The $\theta$-axis is broken for visual clarity.

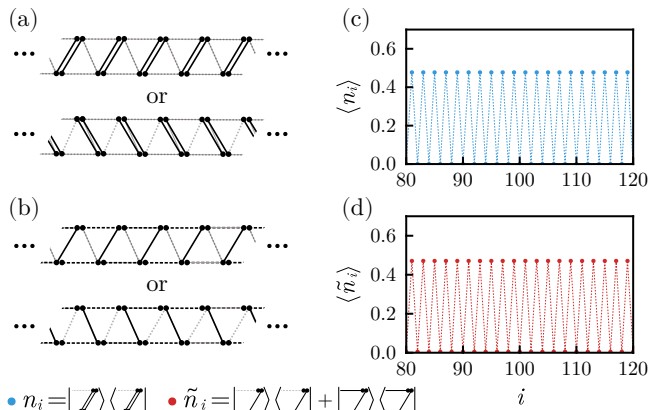

Figure 3: Sketches of two possible $\mathbb{Z}_2$ ordered states that appear as a superposition in the rung-dimerized phase, alongside their typical density profiles. (a) Fully dimerized state with every other rung occupied by a trivial two-dimer loop and (b) partially dimerized state with a single dimer on every other rung. Density profile of two dimers (c) and a single dimer (d) on each rung, computed for the parameters $t' = 0$, $N = 199$ sites, $\delta = -1$ and $\theta = 2$.

the density of columnar dimer states decreases down to zero. Negative $\theta$ stabilizes the $\mathbb{Z}_2$ rung-dimerized phase with spontaneously broken translation symmetry. Although the potential term in the Hamiltonian of Eq.(1) favors only one type of rung-dimerized state - the fully dimerized ones with every other rung occupied by a trivial two-dimer loop - in the $\mathbb{Z}_2$ phase, in actuality these states appear in a superposition with partially dimerized states where every other rung is occupied by a single dimer while the full quantum loop coverage is completed by a uniform density of dimers resonating on the legs of the ladder. In Fig.3 we sketch both pair of states and provide their typical density profiles in the rung-dimerized phase.

After identifying the phases, we take a closer look at the transitions. One way to go from the $\mathbb{Z}_4$ leg-dimerized phase to the disordered phase is through a pair of Ising transitions with an intermediate rung-dimerized phase. At each of the two Ising transitions a $\mathbb{Z}_2$ symmetry is spontaneously broken (hence the $\mathbb{Z}_2$ rung-dimerized phase appears). Upon increasing $\delta$ the two Ising transitions, characterized by the central charge $c = 1/2$, come closer and merge into an Ashkin-Teller multi-critical point with a central charge $c = 1$ [19]. There is some uncertainty in the location of the Ashkin-Teller point due to a limited numerical resolution and a crossover between various critical regimes. Instead, we identify a finite interval along the boundary of the $\mathbb{Z}_4$ leg-dimerized phase where the Ashkin-Teller point is located. Beyond the Ashkin-Teller point the transition becomes chiral. For $t' = t$ we do not observe a floating phase opening up, at least up to $\delta = 50$, indicating that the chiral transition persists for an extremely extended interval.

## 3.2 Ising transitions

We start our analysis with the region where the transition between the $\mathbb{Z}_4$ ordered and the disordered phase takes place through two Ising transitions and an intermediate $\mathbb{Z}_2$ ordered phase. We locate these Ising transitions by performing a finite-size scaling of the relevant order parameter. We first focus on the transition between the $\mathbb{Z}_2$ rung-dimerized and disordered phase. At this transition the translation symmetry between rungs is spontaneously broken. To reflect this broken symmetry, we compute the dimerization on the rungs $D_{\text{rung}} = |\langle n_i \rangle - \langle n_{i+1} \rangle|$, where $n_i = |\,\diagup\,\rangle\langle\,\diagup\,| + 2|\,\diagup\!\diagup\,\rangle\langle\,\diagup\!\diagup\,|$ is an operator that measures the local density of dimers on a rung. In order to reduce edge effects we extract $D_{\text{rung}}$ in the center of the ladder. In the ther-

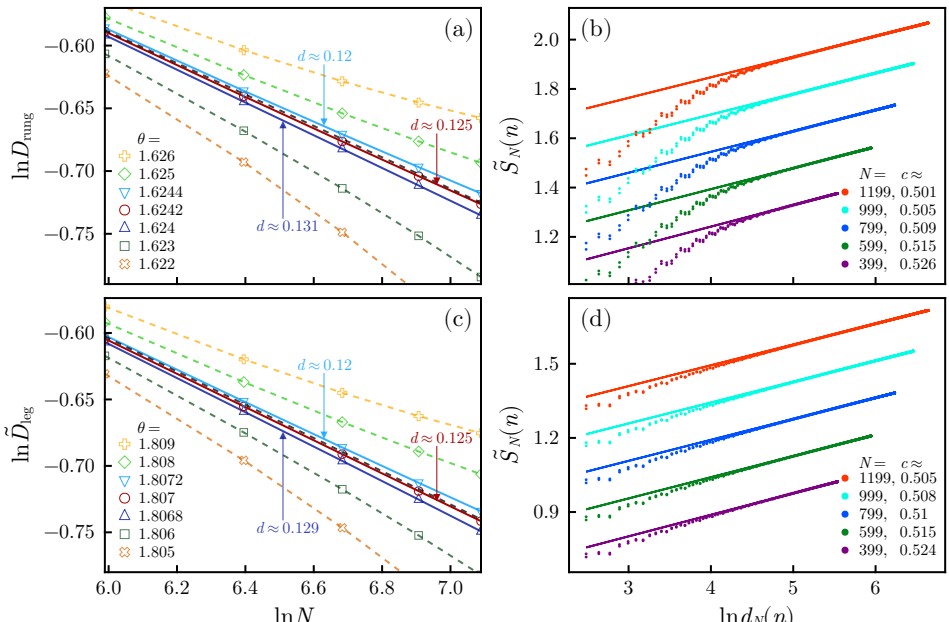

Figure 4: Numerical evidence for the Ising transition between the $\mathbb{Z}_2$ rung-dimerized phase and (a)-(b) the disordered and (c)-(d) the $\mathbb{Z}_4$ leg-dimerized phase for $\delta = 0.55$. (a) Finite-size scaling of the dimerization on the rungs computed in the middle of the ladder with open boundary conditions. The quantum critical point is located at $\theta_1^c \approx 1.6242$ (red circles). (c) Finite size scaling of the dimerization on one of the legs, computed in the middle of the ladder with open boundary conditions. The critical point is located at $\theta_2^c \approx 1.807$ (red circles). In both cases the scaling dimension $d \approx 0.125$ extracted at the critical points is in excellent agreement with theory prediction for Ising transition $d = 1/8$ (black dashed lines). (b),(d) Scaling of the reduced entanglement entropy with $d_N(n) = \frac{2N}{\pi} \sin \frac{\pi n}{N}$ at the two critical points $\theta_1^c$ and $\theta_2^c$ obtained as shown in (a) and (c). Extracted central charges $c \approx 0.501$ at $\theta_1^c$ and $c \approx 0.505$ at $\theta_2^c$ fall within 1% of the $c = 1/2$ of the Ising universality class.

modynamic limit we expect $D_{\mathrm{rung}}$ to approach a finite value inside the ordered $\mathbb{Z}_2$ phase and to vanish in the uniform disordered phase. Consequently, in a log-log plot these will appear as convex and concave curves respectively and in between these we find the separatrix that we associate with the quantum critical point. The slope of the separatrix yields the scaling dimension $d$ of the corresponding operator. In Fig.4 (a) we show the finite-size scaling of the dimerization for $\delta = 0.55$ and various values of $\theta$ in vicinity of the transition. Our numerical results suggest that the critical point is located at $\theta_1^c \approx 1.6242$ and that the numerically extracted scaling dimension $d \approx 0.125$ is in excellent agreement with the conformal field theory predictions $d = 1/8$ for an Ising transition.

To further verify the nature of the transition between the rung-dimerized and disordered phase we extract the central charge $c$. From the reduced density matrix we compute the entanglement entropy $S_N(n)$ and then compute the reduced entanglement entropy by removing Friedel oscillations of the local density [54] - in the present case translation symmetry is broken - which we define as:

$$\tilde{S}_N(n) = S_N(n) - \alpha \langle n_i \rangle, \tag{5}$$

where $\alpha$ is a non-universal parameter tuned such that the oscillations are removed. For conformal transitions the reduced entanglement entropy in 1D systems with open boundary con-

ditions scales linearly with $c$ according to the Calabrese-Cardy formula [55]:

$$\tilde{S}_N = \frac{c}{6} \ln d_N(n) + \ln g + s_1 \,, \tag{6}$$

where $d_N(n) = \frac{2N}{\pi} \sin \frac{\pi n}{N}$ is the conformal distance, $\ln g$ accounts for boundary contributions to the entropy and $s_1$ is a non-universal constant. We extract the central charge at the previously identified critical point $\theta_1^c$ by performing a scaling of the reduced entanglement entropy $\tilde{S}_N(n)$ with the conformal distance as shown in Fig.4 (b). To reduce boundary effects we discard 30% of all sites on both edges of the chain. We observe that under increasing $N$ the numerically extracted $c$ approaches the theory prediction $c = 1/2$ for the Ising transition.

Based on symmetry arguments we also expect an Ising transition between the rung-dimerized ($\mathbb{Z}_2$) and leg-dimerized ($\mathbb{Z}_4$) phase. To confirm this, we follow the same protocol. First, we compute the finite-size scaling of the dimerization, which we denote as $\tilde{D}_{\text{leg}} = |\tilde{n}_i - \tilde{n}_{i+1}|$, in the middle of the ladder, but now for the operator $\tilde{n}_i = |\nearrow\rangle\langle\nearrow| + 2|\rightarrow\rangle\langle\rightarrow|$ that computes the density of dimers on the legs. Here we only consider the top chain, but, of course, the results for the bottom one would be identical. In the rung-dimerized phase the density of dimers on the legs is small and uniform. Consequently, in the thermodynamic limit, $\tilde{D}_{\text{leg}}$ scales to zero in the rung-dimerized phase, while it naturally assumes a finite-value in the leg-dimerized phase. In Fig.4 (c) we provide an example of the scaling of $\tilde{D}_{\text{leg}}$. We use the same value of $\delta = 0.55$ as in Fig.4 (a). The second critical point associated with the separatrix is located at $\theta_2^c \approx 1.807$ and it is clearly different from the previous Ising transition ($\theta_1^c \neq \theta_2^c$). At this point we extract a scaling dimension $d \approx 0.125$ (Fig.4 (c)) and a central charge $c \approx 0.505$ (Fig.4 (d)). Once again, both are in excellent agreement with the Ising universality class.

## 3.3 Ashkin-Teller point and extended chiral transition

By accurately locating the two Ising transitions at the boundary of the $\mathbb{Z}_2$ phase we notice that upon increasing $\delta$ the two transitions come closer and, eventually, the rung-dimerized phase disappears. We expect the multi-critical point where two Ising transitions meet to be in the Ashkin-Teller universality class [18], characterized by a central charge $c = 1$. We also locate the disorder line beyond which the disordered phase has incommensurate short-range correlations. Due to a finite resolution of the numerical approach there is an uncertainty in the location of the multi-critical point and the disorder line. We thus define an interval (indicated in Fig.2 with light pink symbols) that starts once we no longer are able to resolve two Ising transitions with sufficient accuracy and terminates at the point where we detect a chiral transition. The multi-critical Ashkin-Teller point is located within this interval,[3] but there are three possible scenarios that can be realized, as sketched in Fig.5. The most probable situation is that the disorder line hits the Ashkin-Teller point and, if the critical exponent at this point lies within the interval $0.8 \lesssim \nu \lesssim 0.68$, the chiral transition starts immediately, as shown in Fig.5(a). In case when the critical exponent is $0.68 \lesssim \nu \leq 2/3$ the chiral transition starts after a short interval of the conformal Ashkin-Teller transition [22] (see Fig.5(b)). It is also possible that the disorder line hits the transition at a certain distance from the multi-critical point. Then, in the interval between the two, the transition will be commensurate and in the Ashkin-Teller universality class.

Beyond the multi-critical Ashkin-Teller point the transition between the $\mathbb{Z}_4$ and the NNN-Haldane phase can be in one of the three regimes — Ashkin-Teller, chiral, or an intermediate floating phase. Following Huse and Fisher [6], we use the product $\Delta q \times \xi$ to distinguish the critical regimes, where $\Delta q = |q/\pi - 1/2|$ is the distance between the wave-vector $q$ from its

---

[3]We observe a crossover between various critical regimes in the vicinity of the multi-critical point. We therefore associate the Ashkin-Teller multi-critical point with the location that shows the smallest finite-size corrections to the expected Ashkin-Teller critical scaling.

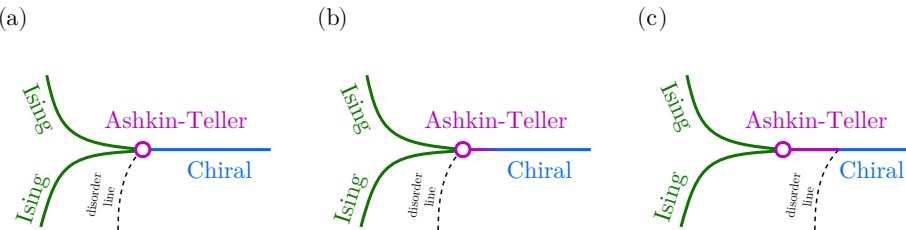

Figure 5: Sketches of the three possible scenarios of two Ising transitions merging into an Ashkin-Teller point followed up by a chiral transition. (a) Disorder line hits an Ashkin-Teller point characterized by $\nu \geq (1 + \sqrt{3})/4 \approx 0.68$. The transition immediately becomes chiral beyond this point. (b) If $2/3 \leq \nu \lesssim 0.68$ the chiral perturbation can be irrelevant, allowing for a short Ashkin-Teller interval, even if the disorder line terminates in the Ashkin-Teller point. (c) The disorder line does not terminate in the Ashkin-Teller points but at the direct transition line, allowing for an Ashkin-Teller interval, even if $\nu \gtrsim 0.68$. Beyond the disorder line, in the presence of chiral perturbations, the transition becomes chiral.

commensurate value $q = \pi/2$ and $\xi$ is the correlation length. Upon approaching the transition $\Delta q$ vanishes with the critical exponent $\bar{\beta}$, while $\xi$ diverges with the critical exponent $\nu$. Despite $\bar{\beta}$ not being known for the Ashkin-Teller universality class, for conformal transitions one expects $\bar{\beta} > \nu$ [3, 6]. Subsequently, the product $|q/\pi - 1/2| \times \xi$ is expected to vanish approaching an Ashkin-Teller transition. For the chiral transition $\bar{\beta} = \nu$ and thus the product $|q/\pi - 1/2| \times \xi$ converges to a constant. By contract, the floating phase, being incommensurate in nature, is separated from the disordered phase by a Kosterlitz-Thouless transition that is characterized by an exponential divergence of the correlation length [12] while at the same time the wave-vector $q$ remains incommensurate through the transition. Therefore, the product $|q/\pi - 1/2| \times \xi$ is expected to diverge in this case.

Our numerical results are summarized in Fig.6, where we show the inverse of the correlation length $1/\xi$, the distance between the wave-vector $q$ from its commensurate value $|q/\pi - 1/2|$, and the product of the two, $|q/\pi - 1/2| \times \xi$. The results are presented for three vertical cuts across the transition for various values of $\delta$. For $\delta = 1.7$, where we expect to go through the Ashkin-Teller point, we extract a critical exponent $\nu \approx 0.678$ and $\bar{\beta} > \nu$, consistent with the Ashkin-Teller universality class. Furthermore, the product $|q/\pi - 1/2| \times \xi$ vanishes upon approaching the transition. In principle, this Ashkin-Teller point belongs to the interval in which chiral perturbations can be irrelevant, i.e. $2/3 \leq \nu \leq (1 + \sqrt{3})/4$ [2, 22], and thus there might be a small interval of the Ashkin-Teller transition, as sketched in Fig.5(b). Our numerical accuracy does not allow to resolve this in more detail. Starting from $\delta \approx 2$ we see signatures of a chiral transition. A typical example of this is presented in Fig.6(b)(e),(h), in which $\delta = 5$. Numerically extracted critical exponents $\nu \approx \bar{\beta} \approx 0.64 \pm 0.03$ are very similar and the product $|q/\pi - 1/2| \times \xi$ stays constant upon approaching the transition, signaling a chiral transition.

Further away from the Ashkin-Teller point we expect the floating phase to open up. Rather surprisingly however, we find that in this quantum loop model the chiral transition extends over a very long interval. Even at $\delta = 50$ (see Fig.6 (c),(f),(i)), the transition still appears to be chiral. If the floating phase appear, the transition that separates it from the $\mathbb{Z}_4$ ordered phase is expected to be in the Pokrovsky-Talapov [11, 56] universality class characterized by the critical exponents $\nu = 1/2$ (correlation length in the ordered phase), and $\bar{\beta} = 1/2$ (incommensurability as an order parameter in the floating phase). The critical exponents that we

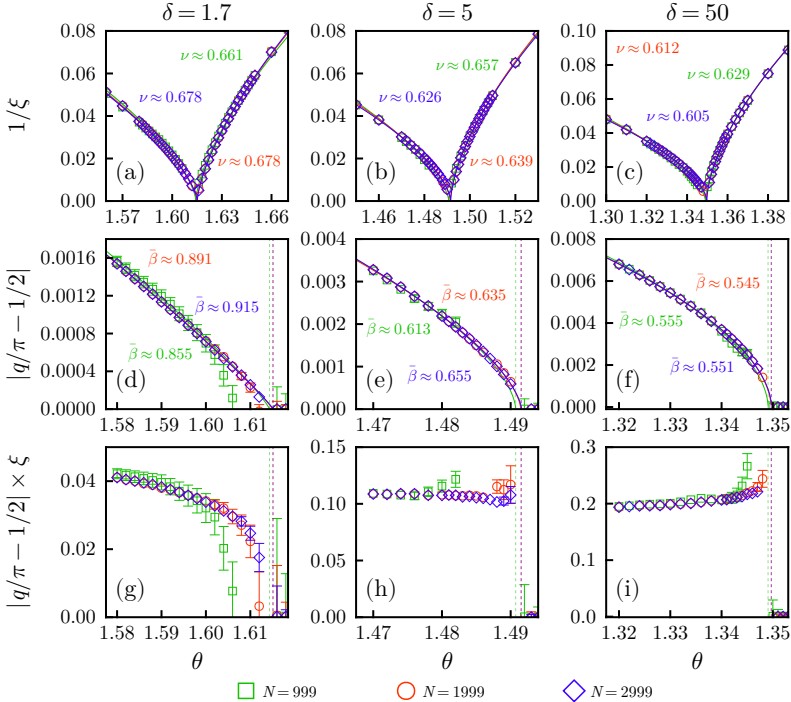

Figure 6: (a)-(c) Inverse of the correlation length $\xi$, (d)-(f) $|q/\pi - 1/2|$ - the distance of wave-vector $q$ with respect to the commensurate value $q = \pi/2$ - and (g)-(i) the product $|q/\pi - 1/2| \times \xi$ along three cuts across the transition between the leg-dimerized ($\mathbb{Z}_4$) and the NNN-Haldane (disordered) phases. We fit $1/\xi$ and $|q/\pi - 1/2|$ with power laws to extract $\nu$ and $\bar{\beta}$. Dashed vertical lines show the boundary of the $\mathbb{Z}_4$ phase extracted by fitting $1/\xi$. Error bars of $|q/\pi - 1/2|$ and $|q/\pi - 1/2| \times \xi$ are shown as $2 \times \xi/N^2$ and $2 \times \xi^2/N^2$ respectively (we only show those that exceed the size of the symbols).

extract numerically across the transition at $\delta = 50$ are still significantly far from these values. Furthermore, $|q/\pi - 1/2| \times \xi$ does not diverge either as expected for the floating phase. In contrary, under increasing system the product $|q/\pi - 1/2| \times \xi$, in the vicinity of the transition, flattens out and clearly converges to a constant, as in agreement with a chiral transition. A remarkably extended interval of a chiral transition in quantum loop models is interesting and inspiring, as the biggest challenge of chiral transitions in the context of Rydberg arrays is their extremely small interval [27, 28].

## 4 Manipulating the nature of $\mathbb{Z}_4$ transitions

In the previous section we presented the phase diagram of the QLM for the special case $t' = t = 1$. Let us now investigate, by changing the value of $t'$, how the nature of the $\mathbb{Z}_4$ transition is altered. In this section we will focus on three cases: first we consider $t' = 0$ associated with an additional fragmentation of the Hilbert space, for which we observe a floating phase; then we present the results for $t' = 2$ where we see no signature of a chiral transition; finally, we fix $\delta = 50$ and study the phase diagram as a function of $\theta$ and $t'$ to track how the nature of the $\mathbb{Z}_4$ transition changes.

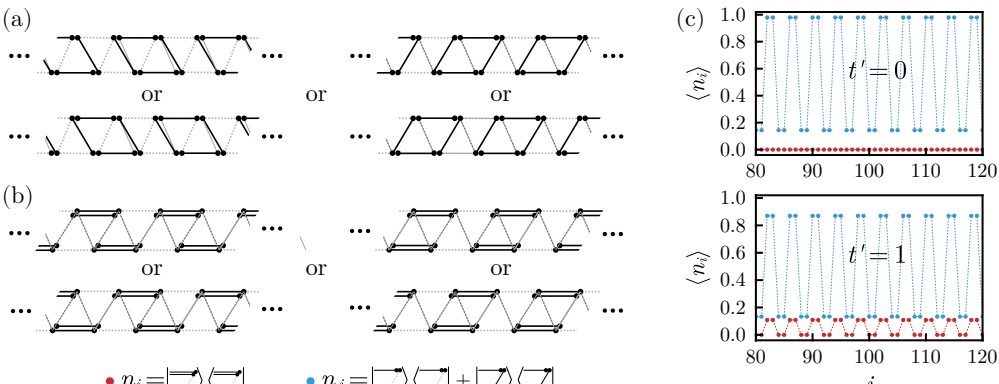

Figure 7: Sketches and density profiles of the $\mathbb{Z}_4$ leg-dimerized phase. (a) Four possible configurations of the plaquette states. (b) The four possible states of the columnar leg phase. (c) Local density of the plaquette (blue) and columnar leg (red) states inside the $\mathbb{Z}_4$ phase for $t' = 0$ and $t' = 1$. We present only the central part of the ladder for $N = 199$, $\delta = 50$ and respectively $\theta = 1.9$ and $\theta = 1.4$.

## 4.1 Transition to the plaquette phase at $t' = 0$

We start our analysis with $t' = 0$ for which the $\mathbb{Z}_4$ columnar leg dimerized states are completely disconnected from other sectors in the Hilbert space. As a result, the $\mathbb{Z}_4$ ordered phase consists solely out of the plaquette phase, like demonstrated in Fig.7. Apart from that, the main features of the phase diagram presented in Fig.8(a) resemble, to a certain extent, the previous case with $t' = 1$. We also detect a $\mathbb{Z}_2$ rung-dimerized phase that is separated from both the $\mathbb{Z}_4$ plaquette

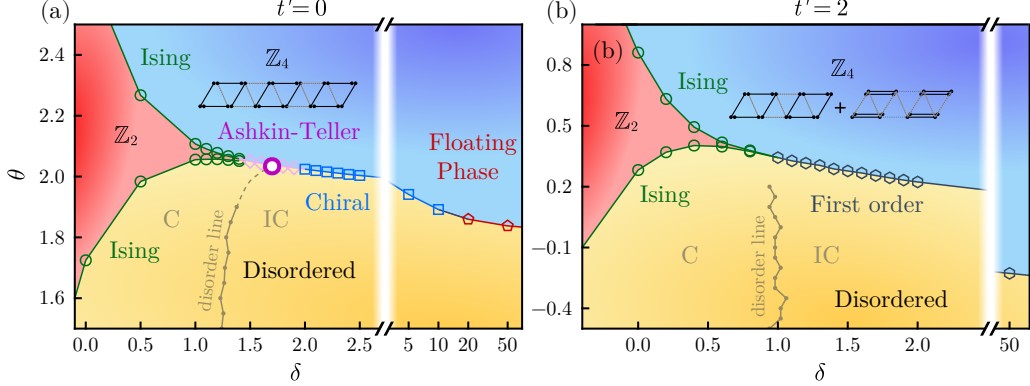

Figure 8: Phase diagrams of the QLM defined by Eq.(1) for (a) $t' = 0$ and (b) $t' = 2$. In each case there are three gapped phases: disordered NNN-Haldane, $\mathbb{Z}_2$ rung-dimerized and $\mathbb{Z}_4$ leg-dimerized. For $t' = 0$ columnar leg states are fully disconnected from the rest of the Hilbert space, causing the $\mathbb{Z}_4$ ordered phase to consist out of the plaquette phase only. In both cases the $\mathbb{Z}_2$ ordered phase is separated from the disordered and the $\mathbb{Z}_4$ ordered phases by continuous Ising transitions (green circles). (a) For $t' = 0$ two Ising transitions meet at the multi-critical Ashkin-Teller point (purple circle), beyond which the transition is chiral (blue squares) until eventually it turns into a floating phase (red pentagons). There is some uncertainty in the location of the Ashkin-Teller point, which we indicate with light pink diamonds. (b) For $t' = 2$, as soon as the $\mathbb{Z}_2$ phase disappears the transition between the disordered and $\mathbb{Z}_4$ phases is first order (dark gray hexagon).

and disordered phase by a pair of Ising transitions. Once again, the rung-dimerized phase disappears and the two Ising transitions meet at the multi-critical Ashkin-Teller point that is followed by the chiral transition. However, this time, the properties of the Ashkin-Teller point and, in turn, the length of the chiral transition are different.

To study the details of the Ashkin-Teller multi-critical point and the transition beyond it, we follow the same protocols as before and compute the scaling of the inverse of the correlation length $1/\xi$ and the product $|q/\pi - 1/2| \times \xi$ for various cuts along the transition line. A few typical examples for $t' = 0$ are presented in Fig.9 (a)-(d). Along the cut at $\delta = 1.8$, that crosses the Ashkin-Teller critical point[4] we see the product $|q/\pi - 1/2| \times \xi$ vanishing (see Fig.9 (b)), in agreement with the conformal transition, and we extract a critical exponent $\nu \approx 0.71$ that is noticeably larger than in the previous case, as demonstrated in Fig.9 (a).

Beyond the Ashkin-Teller point we find that the transition is chiral for an extended interval. In Fig.9(c) we provide an example of $|q/\pi - 1/2| \times \xi$ scaling to a finite-value at the transition. Further away from the Ashkin-Teller point, $\delta = 50$ to be precise, we see a signature of a floating phase with a clear divergence of $|q/\pi - 1/2| \times \xi$, as presented in Fig.9(d). Note that in this case the finite-size effect is opposite to the one we observed for the chiral transition in Fig.6 (i) - $|q/\pi - 1/2| \times \xi$ divergence becomes more apparent under increasing system size.

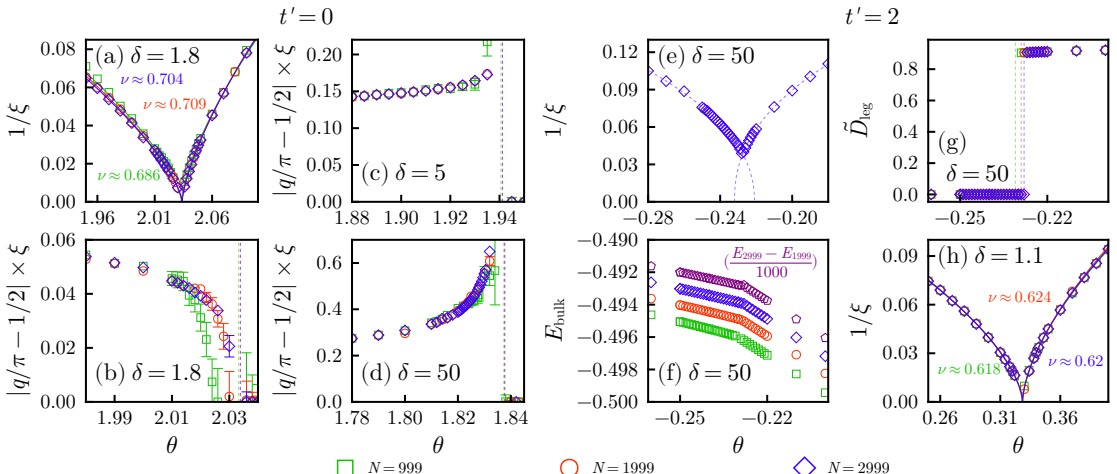

Figure 9: Numerical evidences of various types of quantum phase transition between the $\mathbb{Z}_4$ ordered and disordered phase for (a)-(d) $t' = 0$ and (e)-(g) $t' = 2$. (a) Inverse of the correlation length $1/\xi$ at the Ashkin-Teller point. (b)-(d) The product $|q/\pi - 1/2| \times \xi$ across (a) the conformal Ashkin-Teller transition, (b) direct chiral transition, and (c) the floating phase. Dashed vertical lines indicate the location of the critical point extracted by fitting $1/\xi$. Error bars of $|q/\pi - 1/2|$ are estimated as $2 \times \xi^2/N^2$ and are only shown if they exceed the size of the symbols. (e)-(g) Numerical evidences of the first order transition at $\delta = 50$ that includes (e) finite correlation length $\xi$ across the transition; (f) a kink in the ground-state energy per site, and (g) jump in the order parameter. Dashed lines in (e) indicate a power-law fit. (f) In addition to the finite-size energy per site $E_N/N$ we also compute an estimate of the bulk energy as $(E_{N_1} - E_{N_2})/(N_1 - N_2)$ (purple pentagons) that has significantly reduces boundary effects. (g) Vertical dashed lines indicate the location of the transitions. (h) Scaling of the $1/\xi$ for $t' = 2$ through the multi-critical point. Although the transition appears continuous, the critical exponent $\nu$ is outside the range of the Ashkin-Teller model and might signal a weak first order transition.

---

[4]As in the previous case, there is some uncertainty in the exact location of the Ashkin-Teller point.

According to our data the floating phase opens up between $\delta = 10$ and $\delta = 20$, implying that the chiral transition for $t' = 0$ is shorter than the one for $t' = 1$ (though it is still remarkably extended). This observation fully agrees with the theory prediction that if the Ashkin-Teller conformal point becomes closer to the point with $\nu \approx 0.8$ where it crosses the Lifshitz line, the interval of the chiral transition shortens [20, 22].

To summarize, we find that the nature of the multi-critical Ashkin-Teller point, and in turn the length of the chiral transition, can be manipulated by $t'$, which is responsible for the appearance of double dimers on the legs of the ladder. More precisely, we observe that by switching from $t' = 0$ to $t' = 1$ the critical exponent $\nu$ decays from $\nu \approx 0.71$ to $\nu \approx 0.68$ accompanied by a significant increase of the interval of chiral transition. However, one can also notice that at $t' = 1$ the Ashkin-Teller point is already pretty close to the four-state Potts point with $\nu = 2/3$ [18] and it is therefore natural to question the fate of the chiral transition if we increase $t'$ even further.

## 4.2 First order transition for $t' = 2$

By increasing $t'$ the density of double dimers on the legs in the $\mathbb{Z}_4$ phase is increasing. As a result the $\mathbb{Z}_4$ phase appears earlier and the transition takes place at smaller $\theta$. The phase diagram for $t' = 2$ is presented in Fig.8(b). Like before we observe the rung-dimerized $\mathbb{Z}_2$ phase to be separated by a pair of Ising transitions from the disordered NNN-Haldane phase on one side and the $\mathbb{Z}_4$ leg-dimerized phase on the other. However, quite different from the previous two cases, starting from the point where two Ising transitions meet the two phases are separated by a first order transition. In Fig.9(e)-(g) we present numerical evidences for $\delta = 50$ supporting this claim. First, the correlation length $\xi$ does not diverge at the transition as shown in Fig.9(e). Second, the ground-state energy per site $E_N/N$ has a kink typical for a first order transition that becomes more pronounced upon approaching the thermodynamic limit, as demonstrated in Fig.9(f). In addition to the finite-size results for $E_N/N$ we also consider the bulk energy per site $(E_{N_1} - E_{N_2})/(N_1 - N_2)$ for two different system sizes, effectively eliminating boundary effects, and observe a kink as well. Finally, in Fig.9(g) we show that the dimerization on the legs $\tilde{D}_{\text{leg}}$ of the top chain - the order parameter of the $\mathbb{Z}_4$ leg-dimerized phase - has a clear jump at the transition.

Upon approaching the multi-critical point signatures of the first order transition weaken. In particular, the scaling of the inverse of the correlation length presented in Fig.9(h) looks very similar to that of a continuous transition. However, the numerically extracted critical exponent $\nu \approx 0.62$ lies outside the range $2/3 \leq \nu \leq 1$ defined by the Ashkin-Teller critical theory, with the lowest value $\nu = 2/3$ corresponding to the 4-state Potts point. So, what is the nature of the transition beyond this point? A recent study of the Ashkin-Teller model has shown that the system along the self-dual line becomes gapped for $\nu < 2/3$ [21]. This is compatible with the first order transition that we observe, with a finite correlation length and co-existing ground-states.

## 4.3 Phase diagram as a function of $t'$

For the three examples we considered thus far, i.e. $t' = 0, 1$ and $2$, we observe that for large $\delta$ the transition out of the $\mathbb{Z}_4$ ordered phase can either be through an intermediate floating phase ($t' = 0$), direct and chiral ($t' = 1$), or first order ($t' = 2$). In this section we study how one type of the transition changes into another as a function of $t'$ and take a closer look on how the first order transition opens up.

Since the floating phase appears only for very large values of $\delta$, we focus on $\delta = 50$, which is also sufficiently far to reduce a crossover effect from the multi-critical point where the rung-dimerized phase disappears. Fig.10(a) presents the ground-state phase diagram, as a function

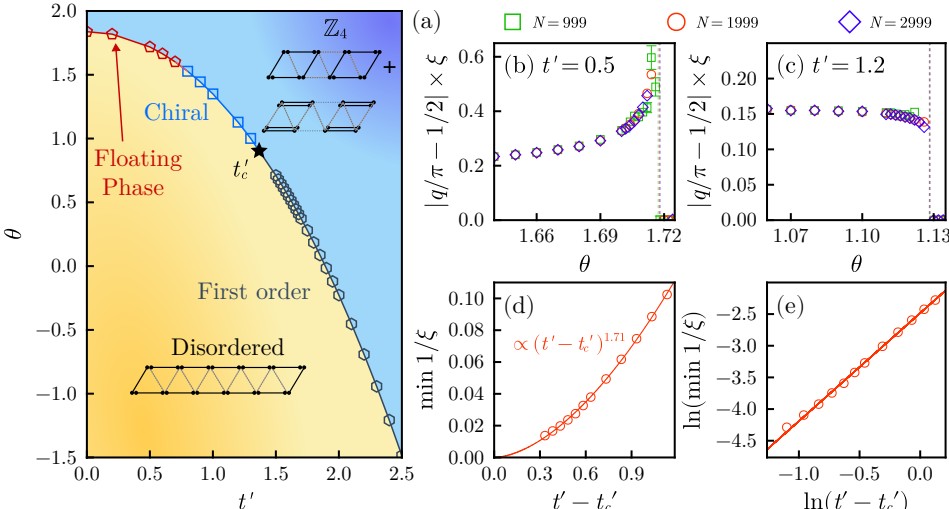

Figure 10: (a) Phase diagram of the QLM as a function of $t'$ and $\theta$ for $\delta = 50$. $t'_c \approx 1.37$ indicates the end point of the chiral transition (black star). (b) and (c) Product $|q/\pi - 1/2| \times \xi$ for a vertical across (b) a floating phase and (c) a chiral transition. (d) Minimum in $1/\xi$ as a function of $t'$ along the first order transition. The solid line indicates a power-law fit. (e) Same results, but but in a log-log scale.

of $\theta$ and $t'$, that summarizes our results. The floating phase that we observe for small $t'$ (see Fig.10(b)) turns into a chiral transition (see Fig.10(c)) between $\theta \approx 1.60$ and $t' \approx 0.7$, and $\theta \approx 1.53$ and $t' \approx 0.8$. In between these points we expect the multi-critical Lifshitz point, with dynamical critical exponent $z = 3$, to be located. Determining the precise location of this point is an extremely challenging computational task and is beyond the scope of this paper. Then, starting from $t'_c \approx 1.37$ we detect a first order transition with finite correlation length at the transition.

By taking a closer look at the minimum in $1/\xi$ we study how the characteristic length scale develops along the first order line and away from the end point of the chiral transition.[5] We identify the location of the transition and the minimum of $1/\xi$ by fitting both sides of the transition with a power law function and calculating the intersection of these fits. In Fig.10(d) we show how the characteristic length develops along the transition. We fit the data point with a power-law, which allows us to estimate the location of the critical point $t'_c$ and its scaling. In Fig.10(e) we present the same set of data but in a log-log scale.[6]

As a final remark let us mention that we observe a clear coexistence of two domains in the disordered phase, another clear signature of a first order transition. In Fig.11 we show the density profiles and entanglement entropy at the first order transition for $t\prime = 1.8$. Close to the edges of the chain we observe a plaquette domain while in the bulk it corresponds to the NNN-Haldane phase. Furthermore, the location of the domain walls clearly matches the peaks in the entanglement entropy.

---

[5]For conformal transitions that would be equivalent to looking at the opening of the energy gap along the first order line, away from the end point. But, since the chiral transition is not conformal and the dynamical critical exponent might not even take a universal value along the transition, we cannot fully rely on our characteristic length results to predict the behavior of the energy gap.

[6]We also checked checked whether the scaling of $\min 1/\xi$ with the distance to the end point can be described with an exponential growth, but this was not feasible.

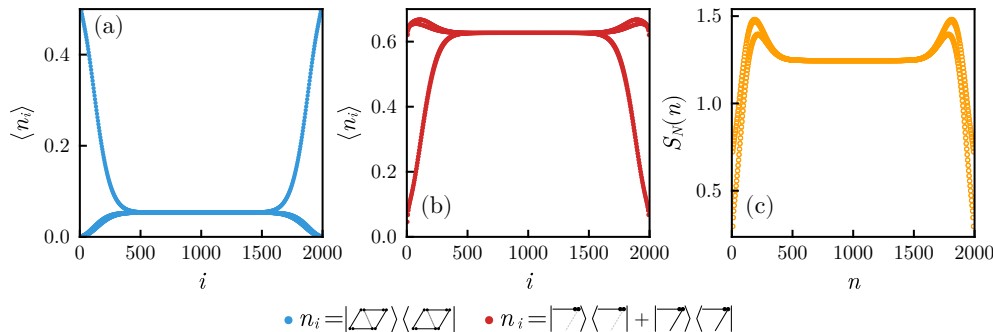

Figure 11: Numerical evidence for the coexistence of two domains in the disordered phase close to the first order transition, for $t' = 1.8$. Density profiles of (a) an entire plaquette and (b) occupation of one dimer on a leg. (c) Entanglement entropy throughout the chain. Quantities are computed for $N = 1999$ sites, $\delta = 50$ and $\theta = 0.184$.

# 5  Conclusion

In this paper we investigate the nature of the transition out of $\mathbb{Z}_4$ leg-dimerized phases in quantum loop models constrained to two dimers per node on a zig-zag ladder. We report very rich critical behavior. First of all, the $\mathbb{Z}_4$ order can be destroyed in two steps via Ising transitions and a rung-dimerized phase - another gapped phase with spontaneously broken $\mathbb{Z}_2$ symmetry. Realization of other scenarios depends on the parameter $t'$, that controls the density of double-dimers on the legs of the ladder. When $t'$ is not too large we observe a multicritical Ashkin-Teller point followed by an extended interval of a $\mathbb{Z}_4$ exotic chiral transition. For small $t'$ the transition can also be through a floating phase - incommensurate Luttinger liquid phase bounded by the Kosterlitz-Thouless and Pokrovsky-Talapov transitions. However, for large values of $t'$ we see no Ashkin-Teller conformal point, nor a chiral transition or a floating phase. Instead, our results predict a first order transition.

The way how this first order transition opens up is actually very interesting. The characteristic length scale diverges algebraically away from the end point of the chiral transition and not exponentially. The latter scenario would suggest the presence of a marginal operator, while the former one, what we actually observe in Fig.10(a), hints towards an additional relevant operator present in the system. As a consequence, we expect the chiral transition to terminate in a critical end point that might have a different underlying critical theory than the continuous chiral transition above it. Numerically this question is very challenging and goes beyond the scope of this work, but we hope that our observation of the chiral transition turning into the first order (to the best of our knowledge it is the first reported example) will advance the development of the field theory of chiral transitions and stimulate further numerical studies in this direction.

Our results obtained for a family of QLMs - a toy model of quantum magnets that completely discards magnetic degrees of freedom - provide a number of important messages to advance the study of $\mathbb{Z}_4$ transitions in more realistic models including, for instance, a Heisenberg spin-1 ladder. First of all, we have confirmed our original conjecture that the VBS transition between leg-dimerized phase and NNN-Haldane phase can be in the $\mathbb{Z}_4$ chiral universality class. The remarkable extent of the chiral transition that we observed significantly increases its chances to be detected in the Heisenberg-like models. Both the plaquette, with a partial dimerization on the legs, and fully-dimerized double-dimer states are present in the $\mathbb{Z}_4$ phase of QLMs - this opens up a wider range of interactions in Heisenberg models to realize the

$\mathbb{Z}_4$ phases. At the same time our results point out that the presence of plaquette states is crucial as strong fully-dimerized state might be responsible for a relevant perturbations that destroys chiral criticality.

Let us also highlight the floating phase, emergent at the boundary of the $\mathbb{Z}_4$ when the plaquette states are dominant. Recently, magnetic floating phases predicted in half-integer $J_1 - J_2$ spin chains attracted a lot of attention among theorists and experimentalists [57–60]. Our results in quantum loop models suggest a non-magnetic counterpart of this exotic phenomena, opening an opportunity to realize them in integer spin-chains.

Furthermore, our results predict that the transition between the $\mathbb{Z}_2$ rung-dimerized and $\mathbb{Z}_4$ leg-dimerized phases, if continuous, will be in the Ising universality class. To the best of our knowledge this possibility has never been reported yet in the context of frustrated Haldane chains. At the same time, the second Ising transition that we observed in QLMs - between the rung-dimerized phase and the NNN-Haldane phase - has been successfully realized in a $J_1 - J_2$ Haldane chain in the presence of biquadratic [36] or three-site [35] interactions. In addition, our result highlight a chance to realize a non-magnetic Ashkin-Teller transition at the critical point where the two Ising transitions meet, of course, under the condition that at this multi-critical point magnetic degrees of freedom would still have a finite gap, which is nevertheless totally feasible.

Finally, let us highlight the peculiarity of the possibility to tune the nature of the multi-critical point, and in turn the length of the chiral transition beyond it, by controlling the density of the double leg-dimerized states with $t'$ in QLMs. Similar mechanisms that allow tuning of the Ashkin-Teller point have recently been proposed in the context of multi-component Rydberg atoms [28]. Despite the two models being remarkably deeply connected, there are some key differences in the fusion rules, and thus also the Hilbert space, between these two models, indicating that the mechanisms responsible for the appearance of the chiral transition in these two models might not be identical (see Appendix C for details). Nevertheless, both families of models - QLMs and an array of multi-component Rydberg atoms - open up an exciting opportunity to manipulate quantum phase transitions.

## Acknowledgments

We thank Andreas Honecker for insightful discussions. NC thanks Yifan Liu for useful comments on the Ashkin-teller model.

**Funding information** This research has been supported by Delft Technology Fellowship. Numerical simulations have been performed at the DelftBlue HPC and at the Dutch national e-infrastructure with the support of the SURF Cooperative.

## A  DMRG with quantum loop model constraints

### A.1  Implementing quantum loop constraints

To fully profit from the fragmented Hilbert space of QLMs, we implement the constraints to which these models are subjected directly into key components of the algorithm, following [48]. The QLMs considered in this study are naturally constrained since each site in the zig-zag ladder forms two, and only two, dimers with other sites. An extra constraint is imposed by limiting the dimer formation to nearest and next-nearest neighbouring sites only. There are several consequences of this. First, instead of considering spins on the lattice as the relevant degrees of freedom we consider the occupation of dimers instead, resulting in the Hilbert space

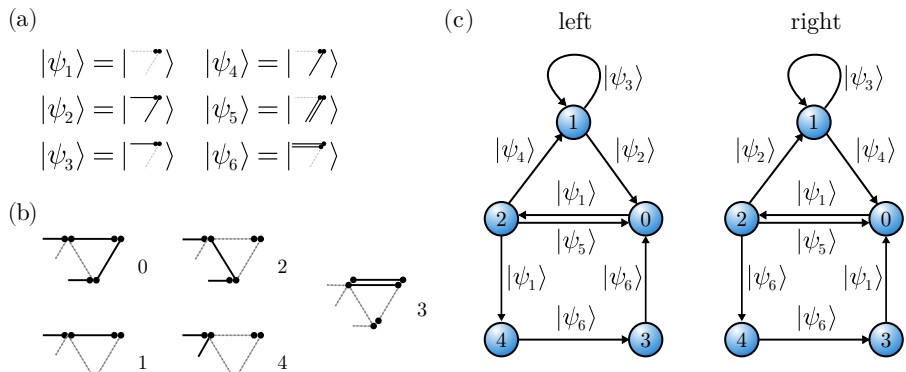

Figure 12: (a) Local Hilbert space of dimension $d = 6$, consisting of all possible dimer occupations on a rung and the preceding leg. (a) Quantum numbers of the left environment that are used to label separate sectors of the Hilbert space. (c) Fusion graphs for left and right environment. There is a one-to-one correspondence between quantum numbers of the left and right.

consisting out of all possible dimer configurations, allowed by the constraints above, on the ladder. One possible local Hilbert space to express the system in is by associating for each site all possible dimer configurations on a single rung and the preceding leg - for $N$ sites in the ladder this translates to $N-1$ QLM degrees of freedom. In total there six of such combinations possible, which we depict in Fig.12 (a). We use these six states as the local Hilbert space for the DMRG algorithm as well.

Second, constraints cause the full Hilbert space to be fragmented into sectors corresponding to different dimer configurations. This fragmentation is not merely a global property but also holds locally during distinct steps of the DMRG algorithm. Take for example the bipartition of the total system into left- and right-normalized blocks — we refer to these as the left and right environments (see [52] for terminology). In the course of sweeping through the chain, one environment growths and the other shrinks. The growth is done iteratively, in which, during each iteration, we add a new layer. In tensor language this corresponds to adding a tensor, and its Hermitian conjugate, of the MPS, and a Matrix Product operator (MPO) tensor to the environment. Each iteration, we label all allowed states of both environments by a set of quantum numbers, corresponding to the different Hilbert space sectors. In total, there are five of such, which we number from 0 to 4. For the left environment these are - for elaborative purposes we use a Valence-Bond-Singlet notation where one spin-1 site is split up into two spin-1/2 dots: 0 has no free dots on either side of the chain, 1 has one free dot on both sides, 2 labels the state in which there are two free dots on the site added, 4 has two free dots on both sides of the chain, and 3 labels the state in which there are two free dots, but such that they do not belong to the last but second last site instead. We depict these quantum numbers of the left environment in Fig.12 (b). A similar set of quantum numbers can be created for the right environment. We note that the labels 0, 1 and 2 are the ones used for the QLM studied in [48] and we added the labels 3 and 4 to allow the formation of the $\mathbb{Z}_4$ columnar phase.

Under the addition of a site to the environment, in dimer language this translates to adding dimers, a quantum number is mapped to a set of new ones. Not all basis are allowed to be added to a given quantum number due to the QLM constraints. Take for example the quantum number zero in which there are no free dots available to form dimers with. As a result we can only add $|\psi_1\rangle$ - the basis state without any dimers - since all other basis states would case the previous spin-1 site in the lattice to form more than two dimers. When the $|\psi_1\rangle$ is added, the environment has two free dots on one side of the chain and is now labeled as 2. By following

the same procedure for the other quantum numbers, we can construct a so called fusion graph that describes the fusion between quantum numbers under the addition of a bases. In the left panel of Fig.12 (c) we list the fusion graph for the left environment. We construct the fusion graph for the right environment (see right panel) by reversing the direction of all arrows and renaming the quantum numbers to be the same as for the left environment. We note that there is a one-to-one correspondence between the Hilbert space sectors of the left and right environment and they can be connected through the following rule: (L, R) = {(0,2), (1,1), (2,0), (3,4), (4,3)}, meaning that the state labeled as 0 on the left can only fuse a state labeled by 2 on the right, etc.

Third, local constraints are also accounted for in determining the optimal tensor(s) and an estimate of the ground state energy at each iteration of the DMRG algorithm. In this process the many-body Hilbert space is mapped to a reduced one in the form of a so-called effective Hamiltonian, which is then diagonalized. The bases in which this eigenvalue problem is greatly reduced in size since it is subjected to the QLM constraints as well, speeding up the diagonalization.

Lastly, splitting up the Hilbert space into distinct sectors allows tensors of the MPS to be written in a block-diagonal form. Not only does this make it so that numerical operations, such as tensor contraction and singular value decomposition, can be carried out on each block separately [61], resulting in them being computationally cheaper. The blocks also require far fewer memory to be stored.

## A.2 Matrix product operator

To use the QLM Hamiltonian shown in Eq.(1) in the constrained DMRG algorithm it first has to be written in terms of the local Hilbert space defined above. By doing so, we obtain the following Hamiltonian

$$\mathcal{H}_{\text{QLM}} = \sum_i (-t p_i^{(1)} p_{i+1}^{(2)} p_{i+2}^{(3)} - t' q_i^{(1)} q_{i+1}^{(2)} q_{i+2}^{(3)} + \text{h.c.}) + \delta r_i - \theta s_i^{(1)} s_{i+1}^{(2)} s_{i+2}^{(3)}, \tag{A.1}$$

where $p^{(1)} = f(4,1)$, $p^{(2)} = f(3,6)$, $p^{(3)} = f(2,6)$, $q^{(1)} = f(5,4) + f(2,3)$, $q^{(2)} = f(1,3)$, $q^{(3)} = f(5,2) + f(4,3)$, $s^{(1)} = f(4,4)$, $s^{(2)} = f(3,3)$, $s^{(3)} = f(2,2)$ and $r = f(5,5)$ and $f(n,m)$ is a $6 \times 6$ matrix for which the element in the $n$'th row and the $m$'th column is 1. To illustrate

$$f(4,1) = \begin{pmatrix} 0 & 0 & 0 & 0 & 0 & 0 \\ 0 & 0 & 0 & 0 & 0 & 0 \\ 0 & 0 & 0 & 0 & 0 & 0 \\ 1 & 0 & 0 & 0 & 0 & 0 \\ 0 & 0 & 0 & 0 & 0 & 0 \\ 0 & 0 & 0 & 0 & 0 & 0 \end{pmatrix}. \tag{A.2}$$

We convert the above Hamiltonian to its MPO form, for which we formulate its tensors

$$\mathcal{W} = \begin{pmatrix}
\mathcal{I} & \cdot & \cdot & \cdot & \cdot & \cdot & \cdot & \cdot & \cdot & \cdot & \cdot & \cdot \\
p^{(3)} & \cdot & \cdot & \cdot & \cdot & \cdot & \cdot & \cdot & \cdot & \cdot & \cdot & \cdot \\
(p^{(3)})^\dagger & \cdot & \cdot & \cdot & \cdot & \cdot & \cdot & \cdot & \cdot & \cdot & \cdot & \cdot \\
q^{(3)} & \cdot & \cdot & \cdot & \cdot & \cdot & \cdot & \cdot & \cdot & \cdot & \cdot & \cdot \\
(q^{(3)})^\dagger & \cdot & \cdot & \cdot & \cdot & \cdot & \cdot & \cdot & \cdot & \cdot & \cdot & \cdot \\
v^{(3)} & \cdot & \cdot & \cdot & \cdot & \cdot & \cdot & \cdot & \cdot & \cdot & \cdot & \cdot \\
\cdot & p^{(2)} & \cdot & \cdot & \cdot & \cdot & \cdot & \cdot & \cdot & \cdot & \cdot & \cdot \\
\cdot & \cdot & (p^{(2)})^\dagger & \cdot & \cdot & \cdot & \cdot & \cdot & \cdot & \cdot & \cdot & \cdot \\
\cdot & \cdot & \cdot & q^{(2)} & \cdot & \cdot & \cdot & \cdot & \cdot & \cdot & \cdot & \cdot \\
\cdot & \cdot & \cdot & \cdot & (q^{(2)})^\dagger & \cdot & \cdot & \cdot & \cdot & \cdot & \cdot & \cdot \\
\cdot & \cdot & \cdot & \cdot & \cdot & v^{(2)} & \cdot & \cdot & \cdot & \cdot & \cdot & \cdot \\
\delta r & \cdot & \cdot & \cdot & \cdot & \cdot & -t p^{(1)} & -t (p^{(1)})^\dagger & -t' q^{(1)} & -t' (q^{(1)})^\dagger & v^{(1)} & \mathcal{I}
\end{pmatrix}, \tag{A.3}$$

with $\cdot$ denoting a $6 \times 6$ matrix with zeros elements only and $\mathcal{I}$ being the $6 \times 6$ identity matrix.

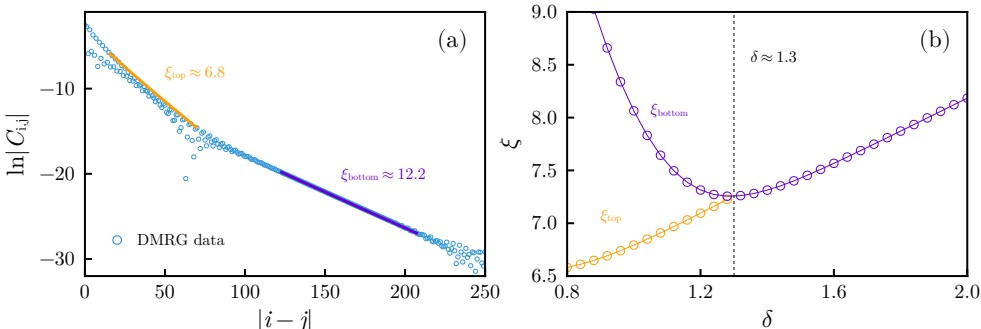

Figure 13: Locating the disorder line via a two step process. (a) First we define a top and bottom segment in the correlation function $C_{i,j}$ in the crossover region. Both of these have their slope fitted to extract their respective correlation lengths $\xi_{\text{top}}$ and $\xi_{\text{bottom}}$. (b) We associate disorder line to be the point where $\xi_{\text{top}} = \xi_{\text{bottom}}$. In the incommensurate domain of the disordered phase we observe only one correlation length.

## B  Locating the disorder line

The disorder line separates the commensurate and incommensurate part of the disordered phase and it is characterized by a kink, frequently a minimum, in the correlation length [62, 63]. Unfortunately, such a kink is not always sharp but often smeared out, making a good estimate of the location of a kink very hard. Therefore, we instead extract an estimate of the disorder line in a two step process, which we illustrate in Fig.13. In vicinity of the disorder line in the commensurate domain, there are two short range orders present in the correlation function; the top part has a smaller correlation length and shows incommensurate correlations while the bottom part reflects the actual order. We denote their respective correlation lengths $\xi_{\text{top}}$ and $\xi_{\text{bottom}}$ (see Fig.13 (a)) and refer to this domain in the phase diagram as the crossover region. In this region $\xi_{\text{top}}$ and $\xi_{\text{bottom}}$ often differ while the in the incommensurate part there is no distinction between the two anymore and there is only a single correlation length. In other words, we associate the point where $\xi_{\text{top}}$ terminates as the estimate of the disorder line, which we define as $\xi_{\text{top}} = \xi_{\text{bottom}}$. We shown an example of this in Fig.13 (b). We note that this is merely an estimate and not the actual disorder line. However, this estimate is consistent with expectations since it approaches the multi-critical Ashkin-Teller point. We are not able to locate the disorder line near this point and leave it out of the data. For indicative purposes we do sketch a possible trajectory to the multi-critical Ashkin-Teller point as a dashed line (see Fig.2 and Fig.8(a))

We locate the disorder for $N = 599$ sites and calculate the correlation function with the operator that measures the density of dimers on the legs $n_i = |\nearrow\rangle\langle\nearrow| + |\nearrow\rangle\langle\nearrow|$. We observe that different operators can result in different estimates of the disorder line, but all of them terminate in, or in vicinity of, the Ashkin-Teller point.

## C  Mapping to multi-component Rydberg atoms

Quantum loop models on zig-zag ladders without states containing two dimes on any legs, such as for $t' = 0$, are rigorously mapped to a Rydberg chain [48]. Let us briefly repeat how. For each rung and the legs it can form a triangle with, i.e. those directly above and below it, we associate a Rydberg atom in the excited state if all three are not occupied by

any dimers. All other possibilities correspond to Rydberg atoms in the ground state. When leg columnar-dimer states are included in the Hilbert space on the other hand, this mapping does not suffice to correctly map the Hilbert space of QLMs to that of Rydberg atoms anymore since this mapping does not a distinguish between the $\mathbb{Z}_4$ ordered plaquette and columnar-dimer phases. To overcome this problem we map the QLM to a multi-component Rydberg chain instead [28, 64, 65].[7] In such a chain, each Rydberg atom can be excited to one of two levels $\alpha = 1, 2$ from the ground state by a laser with Rabi frequency $\Omega_\alpha$ and detuning $\Delta_\alpha$. We use the same mapping as described above to map all states in the Hilbert space without any trivial two-dimer loops on the legs to states with Rydberg atoms that can either be in the ground state or the first excited state. This yields the possibility to describe the NNN-Haldane phase, the $\mathbb{Z}_2$ rung-dimerized phase and the $\mathbb{Z}_4$ plaquette phase. Added to this, we associate for each rung that has both legs it can form a triangle with occupied by two dimers each, a Rydberg atom excited to excitation level. In Fig.14(a) we summarize the above with three illustrative sketches. Additionally, we show the NNN-Haldane phase, rung-dimerized phase and the plaquette and columnar-dimerized phases mapped to a multi-component rydberg chain in Fig.14(b)-(e).

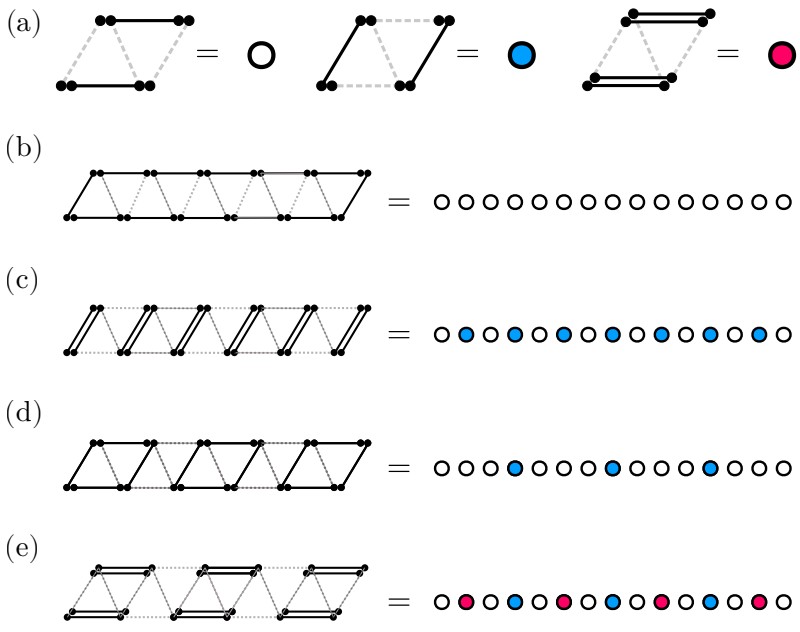

Figure 14: (a) Map between spin-1 quantum loop models, that allow trivial two dimer loops on rungs and legs, and a multi-component Rydberg chain. Shown dimer configurations of the two left sketches are not unique, i.e. there are multiple dimer configurations that map to the same Rydberg state. (b)-(e) Multi-component Rydberg states corresponding to; (b) the NNN-Haldane phase, (c) $\mathbb{Z}_2$ rung-dimerized phase, and the $\mathbb{Z}_4$ (d) plaquette and (e) columnar-dimer phase.

---

[7]In principle, mapping the QLM to a monatomic Rydberg ladder works as well. But, arranging the atoms in such a way that there is a nearest neighbor blockade on the top chain and a three-site blockade on the bottom one, unfortunately seems impossible for larger systems in a 2D geometry.

We derive the constraints to which the Rydberg atoms are subjected to by mapping the QLMs' fusion graph to multi-component Rydberg atoms and describe these in terms of an effective $n$-site blockade model following hard boson statistics. Let us define the operator $c^\dagger_{\alpha,i}$ as the one that excites the $i$'th Rydberg atom from the ground to one of the two excited states. With this in mind, we can formulate the constrains as:

$$n_{1,i}n_{1,i+1} = n_{2,i}n_{2,i+1} = n_{2,i}n_{2,i+2} = n_{1,i}n_{2,i} = n_{1,i}n_{2,i+1} = 0, \tag{C.1}$$

where $n_{\alpha,i} = c^\dagger_{\alpha,i}c_{\alpha,i}$ measure the occupation of a particle in the excitation level $\alpha$ on the $i$'th site.

Using both the Rydberg constraints and the map between QLMs' and Rydbergs' Hilbert space, we map the Hamiltonian (1) to a multi-component Rydberg one as well. We start off with the first term that swaps a pair of dimers. In Rydberg chain, this corresponds to bringing a Rydberg atom from its first excited state to the ground state, such that

$$|\overset{\frown}{\diagup\diagup}\rangle\langle\diagup\diagup| + \text{h.c.} = c_{1,i} + c^\dagger_{1,i}. \tag{C.2}$$

The second kinetic terms maps a plaquette to a leg columnar-dimer one. This, of course, makes it so that we can not simply excite a Rydberg atom from the ground state to the second energy level. To be more precise,

$$|\overset{\frown}{\diagup}\rangle\langle\diagup| + \text{h.c.} = n_{1,i-2}(c_{2,i} + c^\dagger_{2,i})n_{1,i+2}, \tag{C.3}$$

where the occupation number operators ensure that the columnar-dimerized phase can only be mapped to from the plaquette one. Moving on to first the potential term, in QLMs a rung with two dimers on them is preceded and followed by a rung and two legs that are not occupied by any dimers at all. In a Rydberg chain this translates to the Rydberg atoms corresponding to the rung with two dimers on it being in the ground state while its nearest-neighboring atoms are in the first excited state:

$$|\diagup\rangle\langle\diagup| = n_{1,i-1}(1 - n_{1,i})n_{1,i+1} = n_{1,i-1}n_{1,i+1}, \tag{C.4}$$

where the constraint $n_{1,i}n_{1,i-1} = 0$ is used to simplify the expression. Mapping the second potential term is a bit more complex. The three rungs part of the plaquette correspond to Rydberg atoms in the ground state while its neighboring rungs correspond to atoms that are excited to the first energy level. In terms of Rydberg atoms, this is mapped to

$$\begin{aligned}|\diagup\diagup\rangle\langle\diagup\diagup| &= n_{1,i-2}(1 - n_{1,i-1})(1 - n_{1,i})(1 - n_{1,i+1})n_{1,i+2}(1 - n_{2,i})\\ &= n_{1,i-2}(1 - n_{1,i})n_{1,i+2} - n_{1,i-2}n_{2,i}n_{1,i+2}, \end{aligned} \tag{C.5}$$

where again the blockade constraints were used to simplify the expression. The term $(1 - n_{2,i})$ ensures that we actually target the plaquette state and not the leg columnar-dimer one. Despite $n_{1,i-2}n_{1,i+2}$ likely being much smaller than the other potential term - the interaction is distant dependent - it it crucial in stabilizing the plaquette phase. Simply having an energy penalty for the $\mathbb{Z}_2$ and the $\mathbb{Z}_4$ columnar-dimer phase does not ensure that we end up in the plaquette phase - crucial for the realization of the $\mathbb{Z}_4$ chiral transition. Summarizing the above, (1) is mapped to a multi-component Rydberg chain with the Hamiltonian

$$\begin{aligned}\mathcal{H}_{\text{Rydberg}} \approx \sum_i &-t(c_{1,i} + c^\dagger_{1,i}) - t'n_{1,i-2}(c_{2,i} + c^\dagger_{2,i})n_{1,i+2} + (\delta + \theta)n_{1,i}n_{1,i+2}\\ &- \theta n_{1,i}n_{1,i+4} + \theta n_{1,i-2}n_{2,i}n_{1,i+2}. \end{aligned} \tag{C.6}$$

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
