# Peer review of "$\mathbb{Z}_4$ transitions in quantum loop models on a zig-zag ladder"

_SciPost Physics, doi:SciPost Phys. 17, 144 (2024)_

## Round 1 · Referee Report · Anonymous (Referee 4) · 2024-8-3

Strengths
1 - high quality, reliable numerics
2 - provides a novel link between different research areas (frustrated spin chains, quantum loop models, Rydberg atoms)
3 - provides a novel example of a chiral Z_4 transition
Weaknesses
1 - the toy model studied has limited relevance
Report
The paper presents a thorough study of quantum phase transitions in a toy quantum dimer model on a zigzag ladder, inspired by models of frustrated spin-1 chains. It is shown that in an extended parameter range, there is transition between the Z_4 leg-dimerized phase and the "disordered" NNN-Haldane phase, which belongs to the Huse-Fischer chiral universality class. By tuning the model parameters, this transition can be suppressed into 1st order.
The results provide interesting links between different research areas (frustrated spin chains, quantum dimer models, Rydberg atoms) and thus deserve to be published in SciPost.
As a minor shortcoming, one can mention that the realization of the proposed quantum dimer model with the Rydberg atoms on a circular ladder is rather complicated: in addiition to the NNN blockade required for the "inner" circle/leg vs only NN blockade for the outer leg, the constraints require the NN blockade between the inner and outer legs as well; to my opinion, this is not very realistic.
Another minor comment is that it might be confusing that the NNN-Haldane phase is called "disordered" throughout the text, as in possesses a nonlocal order (see Ref.33). It should be remarked that the dimer model considered is constructed in a way that explicitly disfavors the "usual" Haldane phase (which would as well be "disordered" for the purpose of this paper).
Recommendation
Publish (meets expectations and criteria for this Journal)

---

## Round 1 · Referee Report · Anonymous (Referee 3) · 2024-9-9

Strengths
1. detailed and rather thorough study of phase transitions in an effective loop model, which describes the singlet sector of S=1 frustrated chains.
2. impressively large system sizes.
3. rich phase diagram reported, with rather strong evidence regarding the nature of the transitions presented.
Weaknesses
1 . While the finite size effects are shown in some plots, the bond dimension dependence is not presented to the readers.
2. The manuscript text has omissions and the introduction could be made a bit more accessible.
3. While realisation of some the physics of the loop model in experiments is alluded to, it is not highlighted
In my opinion how difficult it will be to measure the nature of the phase transitions in experiments. Can the
authors comment on how realistic it is to measure the critical exponents in Rydberg systems or condensed matter systems ?
Report
This manuscript deals with the quantum loop model description which one of the authors has previously introduced and uses this subspace to analyse a number of quantum phase transitions rather accurately. The methodology is based on dermining the correlation length xi, as well as the oscillation wave vector and its behaviour upon approaching the transition. Based on this information the exponent \nu other aspects of the transitions are inferred. However the dynamical critical exponent is not measured as such.
I think this is a paper of high scientific rigour and should be published in SciPost Physics once the suggested changes have been considered.
Requested changes
1. The notion of loop models is restricted to the authors use in this paper and their previous own work. However loop models have a broader audience, for example as cousins of quantum dimer models. There for example the quantum loop models do not allow for two loop segments to occupy the same link. Please expand the introduction of loop models and clarify the differences with other quantum loop models in quantum magnetism (e.g. square ice ground state manifold)
2. The comment on the maximum bond dimension needs clarification. I am aware of SU(N) models with effective bond dimensions ~10^6, while I would be curious to learn more what allows the present authors to reach 10'000 bond dimension, is this simply brute force, or is there some fragmentation of the total bond dimension into smaller subsection because of the nature of the quantum loop Hilbert space ?
3. Expanding the introduction: As one example I feel that the normal phase of the S=1 chain and its relation to the present problem and its location in the phase diagram would be interesting to know for people who have not yet read the authors previous papers on the subject.
Recommendation
Publish (meets expectations and criteria for this Journal)

---

## Round 2 · Referee Report · Anonymous (Referee 3) · 2024-9-25

Report

I am satisfied with the authors' reaction to all comments raised in the 1st round, and recommend publication.

Recommendation

Publish (meets expectations and criteria for this Journal)

---

## Round 2 · Referee Report · Anonymous (Referee 4) · 2024-10-4

Report

The authors addressed the open points. I can now recommend publication.

Recommendation

Publish (meets expectations and criteria for this Journal)

---

## Round 2 · Author Response

Dear Editor,

We would like to thank both referees for their careful reading of the manuscript and their feedback. We are happy that both recommend publication of the manuscript. Below we address in detail the minor comments raised by the referees. We also attached the revised version of the manuscript.

Yours sincerely,
Bowy La Riviere and Natalia Chepiga

Reply to Referee 1

>>>As a minor shortcoming, one can mention that the realization of the proposed quantum dimer model with the Rydberg atoms on a circular ladder is rather complicated: in addiition to the NNN blockade required for the "inner" circle/leg vs only NN blockade for the outer leg, the constraints require the NN blockade between the inner and outer legs as well; to my opinion, this is not very realistic.

We agree with the referee that acquiring a one-site blockade on the outer chain of the ladder while maintaining a three-site blockade on the inner one through a circular geometry is not very realistic, especially for larger systems. As such, we changed the map of the quantum loop model to that of a multi-component Rydberg chain with two excitation levels per site instead of a circular mono-atomic Rydberg ladder.

>>>Another minor comment is that it might be confusing that the NNN-Haldane phase is called "disordered" throughout the text, as in possesses a nonlocal order (see Ref.33). It should be remarked that the dimer model considered is constructed in a way that explicitly disfavors the "usual" Haldane phase (which would as well be "disordered" for the purpose of this paper).

In principle, any disordered phase can be characterized by a non-local “order parameter”, the correct name for which is the “disorder parameter”. This is the central object in the theory of non-invertible symmetries (dualities) at it maps a disordered phase to an ordered one. For the purpose of this paper it was essential that the disordered phase does not break translation symmetry, has no long range order, has a non-degenerate ground-state and hosts incommensurate short-range correlations. The NNN-Haldane phase satisfies all these criteria.
Now, regarding the differences between the usual Haldane phase and the NNN-Haldane one. The latter is topologically non-trivial, has no protected edge states and thus the domain walls between the NNN-Haldane phase and any of the dimerized or plaquette phases contain no spinor. This results in a non-magnetic transition (Ising, Ashkin-Teller, chiral) taking place within the singlet sector (and therefore they can be captured by the quantum loop model). On the other hand, the Haldane phase is topologically protected and spin-1/2 edge states necessary appear as a spinors at each domain wall, separating Haldane and dimerized/plaquette domains, and resulting in a magnetic transition. The corresponding field theory contains an additional topological term that brings an additional c=1 critical theory on top (e.g. transition into the dimerized phase is WZW SU(2)_2 with c=3/2=1+1/2, where the latter is due to Ising). These transitions, though very interesting, are outside of the scope of this study of quantum loop models.

Reply to Referee 2

>>>The notion of loop models is restricted to the authors use in this paper and their previous own work. However loop models have a broader audience, for example as cousins of quantum dimer models. There for example the quantum loop models do not allow for two loop segments to occupy the same link. Please expand the introduction of loop models and clarify the differences with other quantum loop models in quantum magnetism (e.g. square ice ground state manifold).

We expanded the introduction to clarify the distinction of quantum loop models that we use from some other definitions used in the literature and clarify the connection to the quantum dimer models. Let us also comment that we consider two loop segments occupying the same link as a trivial loop and in this respect our formulation of QLMs is more complete.

>>>The comment on the maximum bond dimension needs clarification. I am aware of SU(N) models with effective bond dimensions ~10^6, while I would be curious to learn more what allows the present authors to reach 10'000 bond dimension, is this simply brute force, or is there some fragmentation of the total bond dimension into smaller subsection because of the nature of the quantum loop Hilbert space ?

There is indeed a fragmentation of the Hilbert space into smaller subsection as a result of the constraints encountered in the construction of states for quantum loop models. Despite this, we noticed in during our simulations that the maximum bond dimension of 10.000 was never reached. We clarified this more clearly in the text.

>>> Expanding the introduction: As one example I feel that the normal phase of the S=1 chain and its relation to the present problem and its location in the phase diagram would be interesting to know for people who have not yet read the authors previous papers on the subject.

We added some more details about the nature of the NNN-Haldane phase, and under which conditions it can be realized, in a spin-1 chain.

---

## Round 2 · List of Changes

- Expanded the introduction to give more details about the nature of the next-nearest-neighbor Haldane phase in quantum spin-1 chains, and how it can be realized.
- Clarified more clearly in the text why the maximum bond dimension of 10.000 was never reached as a result of the fragmented Hilbert space of quantum loop models.
- Expanded the introduction to clarify the distinction between quantum loop models and quantum dimer models.
- Changed the mapping to a circular Rydberg ladder from the studied quantum loop models to a multi-component Rydberg chain with two excitation levels.
- Improved grammar of the text and removed typos.

---

## Editorial Decision

published